# Plasminogen Activator Inhibitor-1 in poorly controlled vs well controlled Type-2 Diabetes Mellitus patients: A case-control study in a district hospital in Ghana

Charles Nkansah[1,2], Otchere Addai-Mensah[1], Kofi Mensah[1,3], Michael Owusu[1], Richard K. D. Ephraim[4], Patrick Adu[4], Felix Osei-Boakye📶[1,5], Samuel K. Appiah[1,6], Dorcas Serwaa📶[7]*, Charles A. Derigubah[1,8], Alexander Yaw Debrah[1]

1 Department of Medical Diagnostics, Faculty of Allied Health Sciences, Kwame Nkrumah University of Science and Technology, Kumasi, Ghana, 2 Clinical Laboratory Department, Nkenkaasu District Hospital, Nkenkaasu, Ghana, 3 Clinical Laboratory Department, Komfo Anokye Teaching Hospital, Kumasi, Ghana, 4 Department of Medical Laboratory Technology, Faculty of Allied Health Sciences, University of Cape Coast, Cape Coast, Ghana, 5 Clinical Laboratory Department, Mankranso District Hospital, Mankranso, Ghana, 6 Clinical Laboratory Department, Wenchi Methodist Hospital, Wenchi, Ghana, 7 Reproductive Biology Unit, Department of Obstetrics and Gynaecology, College of Medicine, Pan African University of Life and Earth Sciences Institute (PAULESI), University of Ibadan, Ibadan, Nigeria, 8 Clinical Laboratory Department, Tamale Teaching Hospital, Tamale, Ghana

* dserwaa0327@stu.ui.edu.ng

**Data Availability Statement:** All relevant data are within the manuscript and its Supporting information files.

## Abstract

### Background

Hypofibrinolysis resulting from the up-regulation of plasminogen activator inhibitor-1 (PAI-1) usually occurs in patients with type 2 diabetes mellitus (T2DM), rendering them hypercoagulable. This study assessed the plasma antigen and activity levels of the PAI-1 enzyme in T2DM patients in a district hospital in Ghana.

### Methods

This was a hospital-based case-control study conducted from December 2018 to May 2019 at Nkenkaasu District Hospital. Sixty subjects with T2DM (30 T2DM subjects with good glycemic control and 30 with poor glycemic control), and 30 apparently healthy blood donors were recruited into the study. Blood specimens were collected for complete blood count, lipid profile, PAI-1 Ag and PAI-1 activity levels. A pre-tested questionnaire was used to obtain demographic and clinical information. The data was analyzed using SPSS version 22.0.

### Results

Elevated PAI-1 Ag and activity levels were observed in the T2DM subjects compared to the healthy controls, with the levels and activity significantly higher (PAI-1 Ag; $p < 0.001$, PAI-1 activity level; $p = 0.004$) in the T2DM subjects with poor glycemic control in comparison to those with good glycemic control. A significant positive correlation was observed between

**Funding:** The author(s) received no specific funding for this work.

**Competing interests:** The authors have declared that no competing interests exist.

HbA1c and PAI-1 enzymes. PAI-1 Ag levels significantly increased along with increased total cholesterol ($B = 0.262$, $p = 0.033$), triglyceride ($B = -0.273$, $p = 0.034$) and HbA1c ($B = 0.419$, $p = 0.001$). Similarly, PAI-1 activity level was associated with total cholesterol ($B = 0.325$, $p = 0.009$), triglyceride ($B = -0.262$, $p = 0.042$), HbA1c ($B = 0.389$, $p = 0.003$) and VLDL-c ($B = -0.227$, $p = 0.029$).

## Conclusion

PAI-1 antigen/activity is enhanced in poorly controlled Ghanaian T2DM subjects. The hypercoagulable state of the affected individuals put them at higher risk of developing cardiovascular diseases. Good glycemic control to regulate plasma PAI-1 levels is essential during T2DM lifelong management. Markers of fibrinolysis should be assessed in these individuals and appropriate anticoagulants given to prevent thrombosis and adverse cardiovascular diseases.

## Introduction

Diabetes mellitus is a chronic metabolic disorder of multifaceted etiology characterised by persistent elevation of plasma glucose levels resulting from the defects in the normal regulation of carbohydrates, fats and proteins in the body [1]. The condition stems primarily from deficiency in insulin secretion, or action, or both and may manifest with symptoms such as frequent thirst, polyuria, polyphagia, blurring of vision and weight loss [2]. Globally, about 450 million people have diabetes, majority (90–95%) of them being type 2 diabetes mellitus (T2DM) with 5.1 million dying from it yearly [3]. The worldwide prevalence of diabetes mellitus is 8.5% [4], and this figure is expected to increase to 17.6% by 2030, excluding the high numbers of undiagnosed cases which is estimated around 175 million [5]. In 2013, Sub-Saharan Africa had 21.5 million people with diabetes and approximately five hundred thousand died from the condition [4]. *Gatimu et al.*, (2016) found the weighted prevalence of diabetes among the adults aged 50 years and above in Ghana to be 3.95%, and 3% in the general Ghanaian population [6].

Type 2 diabetes mellitus (T2DM) participants are more likely than non-diabetics to suffer from thrombotic events [7]. The hypercoagulable state of T2DM patients has recently been confirmed to be resulting from the associated elevated secretion and/or activation of pro-coagulants coupled with hypofibrinolysis [8]. The prothrombotic state is characterized by increased tissue factor [7], factor IV [9], thrombin [10], and fibrinogen levels [11]. The disturbance in fibrinolysis is linked to the subsequent decrease in local tissue plasminogen activator production and elevation in the synthesis and/or activation of plasminogen activator inhibitor-1 (PAI-1), which is the main inhibitor of fibrinolysis [12]. Such elevated levels of PAI-1 have been linked to an increased risk of myocardial infarction (MI), ischemic heart disease (IHD) and cardiovascular disease (CVD) [13].

Regardless of the several interventions to manage T2DM, cardiovascular diseases constituting about 75% to 80%, are still the leading cause of mortality globally in patients with T2DM [14]. The lowered plasma fibrinolytic activity, mainly ascribable to the elevated plasma antigen and activity levels of PAI-1, has been associated with the development of cardiovascular diseases and recurrent myocardial infarction according to a study in 2018 which concluded that good glycemic control during T2DM management is very essential to reduce the risk of

cardiovascular disease onset [15]. The hypercoagulable state in T2DM participants has been confirmed by Ephraim et al., (2017) who observed reduced Activated Partial Thromboplastin Time (APTT) and Prothrombin Time (PT) among T2DM patients in a district hospital in Ghana. Another study in Ghana by Addai-Mensah *et al.*, [16] also found decreased levels and activities of proteins S, C, and antithrombin III in T2DM patients with poor glycemic control, further confirming the high risk of these patients to thrombosis.

Despite the unique role of PAI-1 in hemostasis, as an independent potent marker for thrombosis, this enzyme is barely assessed during the lifelong management of diabetes mellitus patients, especially in sub-Sahara Africa. Meanwhile, failure to detect the altered changes in fibrinolysis early puts T2DM patients at a higher risk of thromboembolism and this may contribute to the increased mortality in patients with diabetes mellitus due to the associated thrombosis and cardiovascular complications [15]. To the best of our knowledge, there has not been a single study in Ghana to assess the plasma levels of PAI-1 among T2DM patients. The study by Ephraim *et al.*, (2017) was limited to only the screening test for coagulation and could not give account on the state of the plasma antigen and activity levels of plasminogen activator inhibitor-1. The Addai-Mensah *et al.*, (2019) study also measured anticoagulation markers and did not assess PAI-1 in type-2 diabetes mellitus patients with poor glycemic control. This study was therefore designed to assess the plasma antigen and activity levels of plasminogen activator inhibitor- 1 (PAI-1) in type-2 diabetes patients in a district hospital in Ghana.

## Participants and methods

### Study site/design

This hospital-based case-control study was conducted from December 2018 to May 2019 at Nkenkaasu District Hospital located in the Offinso-North district, in the Ashanti Region, Ghana with a total population of about 68,543. The Nkenkaasu District Hospital serves as the main referral facility in the district and its neighboring villages. This Hospital records about 472 cases of diabetes annually, with 427 of them being T2DM, (per the outpatient department's report). The total land area of the Offinso-North district is about 945.9 square kilometres and lies between latitude: 7˚20N. 6˚50S" and longitude: 1˚60W", 1˚45E". Majority of the inhabitants of this district are farmers [17].

### Study population

A total of 90 participants: 60 type 2 diabetes mellitus patients: (30 T2DM subjects with good glycemic control and 30 with poor glycemic control) attending the Diabetes Clinic of Nkenkaasu District Hospital in Offinso-North district and 30 apparently healthy blood donors were recruited into this study. T2DM patients with HbA1c value < 7% were said to be experiencing good glycemic control and those with HbA1c value $\geq$7 were considered to be having poor glycemic control [18]. Individuals with type 1 diabetes mellitus, newly diagnosed T2DM patients, participants with known CVD, liver diseases, kidney diseases, coagulopathies, enzymopathies, hemoglobinopathies, hepatitis, HIV and tuberculosis were excluded from the study.

### Sample size determination

The necessary sample size was obtained by employing the Kelsey's formula:

$$N_{cases-Kelsey} = \left[\frac{r+1}{r}\right]\frac{P(1-P)\left(Z_{\frac{\alpha}{2}} + Z_\beta\right)^2}{(p1-p2)^2},$$

and

$$P = \left[ \frac{p1 + (r \ X \ p2)}{r + 1} \right],$$

where r is the ratio of T2DM to healthy controls, which is 2:1 in this study, $Z_{\frac{\alpha}{2}}$ represents the critical value of the normal dispersion at $\alpha/2$ (for this study at confidence level of 95%, $\alpha$ is 0.05 and the critical value is 1.96), $Z_{\beta}$ represents the critical value of the normal distribution at $\beta$ (this study used a power of 80%, $\beta$ is 0.2 and the critical value is 0.84. p1 represents the percentage of the risk of thrombosis in diabetic group, 40% according to Tsai *et al*., (2002), p2 is the proportion of the risk of thrombosis in the control group, 10.7% according to Piazza *et al*., (2012) and p1-p2 is the smallest difference in proportions that is clinically important.

From the formula above, the minimum number of T2DM required for this study was 58 with corresponding controls of 29. However, this study employed 90 subjects: 60 T2DM patients and 30 healthy controls.

## Administration of questionnaire

A well-structured and pre-tested questionnaire was administered to obtain socio-demographic variables and clinical history from the participants.

## Anthropometric variables measurements

**Body mass index.** Height to the nearest centimeter without shoes and weight to the nearest 0.1 kg in light clothing was estimated. Participants were weighed on a bathroom scale and their heights were measured with a wall-mounted ruler. Body mass index (BMI) was calculated by dividing weight (kg) by height squared ($m^2$). BMI was categorized as: <18.5 (underweight); 18.5 to 24.5 (normal weight); 25 to 29.5 (overweight); and ≥30 (obese) [2].

**Blood pressure (using Korotkoff 1 and 5).** Blood pressure was measured by trained personnel using a mercury sphygmomanometer and a stethoscope. Measurements were taken from the left upper arm after participants sat >5 min in accordance with the recommendations of the American Heart Association [19]. Duplicate measurements were taken with a 5-minutes rest interval between measurements and the mean value was recorded in mmHg. Hypertension was graded as normal when the systolic blood pressure (SBP) was >120 mm Hg and diastolic blood pressure (DBP) >80 mm Hg [2].

**Blood sample collection.** Ten (10) mL of venous blood was collected aseptically from each participant between 8.00 and 10.00 a.m.; 3.4 mL into EDTA for HbA1c; 3.6 mL dispensed into 3.8% sodium citrate for PAI-1 antigen and activity levels; and 3ml into gel tube for the glucose estimation. Plasma was separated after centrifugation (CENTRIFUGE 80–1, Japan) and analyzed immediately and part aliquoted into eppendorf tubes and stored at -20˚C until analysis.

**Laboratory assays.** All laboratory investigations were done at the Methodist Hospital Laboratory, Wenchi, Bono Region, Ghana. Glycated hemoglobin (HbA1c) was done using Microlab 300, Vital Scientific, from Japan with reagents from Medsource Ozone Biomedicals Pvt. Ltd., Japan. The fasting plasma glucose estimated using Mindray: BS-200E with reagents from ELITech Clinical Systems. PAI-1 Ag and PAI-1 activity levels were determined by an enzyme-linked immunosorbent assay (ELISA) (Biobase, China) using automated ELISA washer (BIO-RAD, PW40) and ELISA reader (Mindray, MR-96A).

### Ethical consideration and informed consent

Ethical approval for the study was obtained from the Committee on Human Research, Publication and Ethics of the Kwame Nkrumah University of Science and Technology (CHRPE/AP/ 300/19) and the hospital authorities.

Written informed consent of individual participants was sought after the aims and objectives of the study had been thoroughly explained to them. Participants either signed or thumbprinted to give their consent, before the commencement of the study and they assured of the confidentiality of their data.

### Statistical analysis

Data obtained were analyzed with Statistical Package for the Social Sciences (SPSS), version 22.0. Test for normality was done with box plot and Kolmogorov-Smirnoff test. Parametric data and non-parametric data were presented as means ± standard deviation and median (interquartile ranges) respectively. Frequencies and percentages were calculated to enable comparison of characteristics between T2DM subjects with variable glycemic control and normal controls. The Chi square ($X^2$) or Fishers exact test was used appropriately to test the descriptive statistics for the categorical variables and the Student's T test or the Mann–Whitney U test whenever applicable used for the continuous variables. Relationships were assessed with Spearman rank test. Continuous variables within the three groups (T2DM subjects with good glycemic control, T2DM participants with poor glycemic control and healthy controls) were compared using the Kruskal Wallis analysis of variance. A $p$ value of $<0.05$ was considered statistically significant.

## Results

### Demographic, anthropometric and clinical characteristics of the participants stratified by states of glycemic control

General demographic, anthropometric and clinical characteristics of the study population stratified by states of glycemic control. The median age of the participants was 56 (50.75–62.0) years with majority of them above forty years. Sixty (66.7%) out of the 90 participants were females. The median BMI was 25.0 (22.01–28.08) kg/m$^2$. 42.2% had normal weight while 4.4% and 16.7% were underweight or obese respectively. Almost 28% of the study participants did not receive formal education and nearly half of the participants (46.7%) were farmers. Those few T2DM subjects who were alcoholics (2) and smokers (1) had poor glycemic controls, even though this was not statistically significant ($p = 0.129$ and $p = 0.364$ respectively). Sixty per cent (60%) of the T2DM participants had had the condition for less than six years whiles 23.3% had had it for between 6–10 years. Only 5% had had it for more than fifteen years. Age ($p = 0.992$), gender ($p = 0.865$), BMI ($p = 0.344$), formal education ($p = 0.180$), occupation ($p = 0.110$), alcohol consumption ($p = 0.129$) and smoking history ($p = 0.364$) were not statistically significant among the study population (Table 1).

### Clinical, biochemical and hematological characteristics of the participants stratified by the states of glycemic control

Table 2 shows the clinical characteristics of the study population stratified by their states of glycemic control. The median systolic blood pressure (SBP) recorded among the normal controls, 137.00 mmHg (127.75–147.25) was lower compared to the T2DM participants. The highest blood pressure was recorded among those with poor glycemic control (139.50 mmHg (129.50–149.50) and those with good glycemic control also had a median of 138.00 mmHg

**Table 1. Demographic, anthropometric and clinical characteristics of study population stratified by the states of glycemic controls.**

| Variables | T2DM | | Non-Diabetic Subjects | p-value |
|---|---|---|---|---|
| | Good Glycaemic Control | Poor Glycaemic Control | | |
| | N (%)[a] | N (%)[a] | N (%)[a] | |
| **Age group (years)** | | | | |
| <41 | 2 (6.7) | 2 (6.7) | 3 (10.0) | 0.992 |
| 41–50 | 4 (13.3) | 6 (20.0) | 5 (16.7) | |
| 51–60 | 15 (50.0) | 13 (43.3) | 15 (50.0) | |
| 61–70 | 5 (16.7) | 4 (13.3) | 3 (10.0) | |
| >70 | 4 (13.3) | 5 (16.7) | 4 (13.3) | |
| **Gender** | | | | |
| Males | 11 (36.7) | 9 (30.0) | 10 (33.3) | 0.865 |
| females | 19 (63.3) | 21 (70.0) | 20 (66.7) | |
| **BMI ($kg/m^2$)** | | | | |
| Underweight | 1 (3.3) | 3 (10.0) | 0 (0) | 0.344 |
| Normal weight | 10 (33.3) | 12 (40.0) | 16 (53.3) | |
| Overweight | 12 (40.0) | 10 (33.3) | 11 (36.7) | |
| Obese | 7 (23.3) | 5 (16.7) | 3 (10.0) | |
| **Formal education** | | | | |
| Yes | 18 (60.0) | 23 (76.7) | 24 (80.0) | 0.180 |
| No | 12 (40.0) | 7 (23.3) | 6 (20.0) | |
| **Occupation** | | | | |
| Farming | 10 (33.3) | 12 (40.0) | 20 (66.7) | 0.110 |
| Trading | 8 (26.7) | 11 (36.7) | 5 (16.7) | |
| Civil Servants | 5 (16.7) | 2 (6.7) | 3 (10.0) | |
| Unemployed | 7 (23.3) | 5 (16.7) | 2 (6.7) | |
| **Alcohol consumption** | | | | |
| Yes | 0 (0) | 2 (6.7) | 0 (0) | 0.129 |
| No | 30 (100) | 28 (93.3) | 30 (100) | |
| **Smoking** | | | | |
| Yes | 0 (0) | 1 (3.3) | 0 (0) | 0.364 |
| No | 30 (100) | 29 (96.7) | 30 (100) | |
| **T2DM duration (years)** | | | | |
| 0–5 | 22 (73.3) | 14 (46.7) | | 0.001* |
| 6–10 | 4 (13.3) | 10 (33.3) | | |
| 11–15 | 3 (10.0) | 4 (13.3) | N/A | |
| >15 | 1 (3.3) | 2 (6.7) | | |

BMI = Body mass index, T2DM = Type 2 diabetes mellitus.

[a]Data presented as frequencies (N) with corresponding proportions in parentheses.

* p value <0.05

(130.75–147.75). The above observation was similar to the diastolic blood pressure (DBP) which was highest among the participants with poor glycemic control (84.47±15.53) as compared to those with good glycemic control (82.97±10.44) and the normal controls (82.70 ±10.47). The variations in the blood pressures (systolic vs diastolic) among the three groups were not statistically significant (p = 0.800 vs p = 0.838). Fasting plasma glucose was highest among the T2DM patients with poor glycemic control [13.51 mmol/L (12.40–16.43)] compared to those with good glycemic control [7.45 mmol/L (6.01–8.95)]. Glycated hemoglobin

**Table 2. Clinical, biochemical and hematological characteristics of the study population stratified by the states of glycemic control.**

| Variables | T2DM Participants | | Healthy Controls | *p*-value |
|---|---|---|---|---|
| | Good Glycemic Control (HbA1c<7) | Poor Glycemic Control (HbA1c≥7) | | |
| Age (years) | 55.50 (52.50–63.50) | 57.00 (50.00–62.25) | 54.00 (50.00–60.25) | 0.637[a] |
| BMI ($kg/m^2$) | 26.62 (23.15–29.54) | 24.59 (19.48–28.08) | 24.44 (21.98–26.74) | 0.252[a] |
| SBP(mmHg) | 138.00 (130.75–147.75) | 139.50 (129.50–149.50) | 137.00 (127.75–147.25) | 0.800[a] |
| DBP (mmHg) | 82.97 ± 10.44 | 84.47 ± 15.53 | 82.70 ± 10.47 | 0.838[b] |
| FPG (mmol/l) | 7.45 (6.01–8.95) | 13.51 (12.40–16.43) | 6.15 (5.40–6.33) | **≤0.001**[a*] |
| HbA1c (%) | 6.50 (6.20–6.80) | 9.35 (8.60–12.43) | 4.65 (4.30–5.20) | **≤0.001**[a*] |
| Total Cholesterol (mg/dl) | 180.00 (131.25–205.00) | 196.00 (159.75–209.00) | 139.50 (128.00–175.00) | **0.001**[a*] |
| Triglyceride (mg/dl) | 116.00 (80.00–142.00) | 147.50 (129.00–180.50) | 76.50 (47.50–96.00) | **≤0.001**[a*] |
| HDL-c (mg/dl) | 64.00 (55.00–77.75) | 51.60 (38.00–63.00) | 66.39 (45.75–74.73) | **0.002**[a*] |
| LDL-c (mg/dl) | 81.90 (53.30–105.00) | 114.80 (95.68–136.20) | 89.50 (72.00–102.20) | **≤0.001**[a*] |
| VLDL-c (mg/dl) | 23.00 (19.00–28.00) | 26.00 (18.00–28.70) | 22.85 (20.75–26.00) | 0.270[a] |
| Platelet (X10$^9$/l) | 226.50 (178.75–226.75) | 244.00 (193.50–282.00) | 193.00 (178.75–246.00) | 0.163[a] |
| PAI-1 Ag level (ng/ml) | 22.33 (19.72–26.86) | 23.48 (21.16–28.74) | 19.05 (17.82–21.30) | **≤0.001**[a*] |
| PAI-1 Activity (U/ml) | 4.49 (4.06–5.60) | 4.76 (4.19–6.02) | 4.12 (3.44–4.45) | **0.004**[a*] |

BMI = Body mass index, kg/m² = kilogram per meter squared, T2DM = Type 2 diabetes mellitus, SBP = Systolic blood pressure, DBP = Diastolic blood pressure, FPG = Fasting plasma glucose, HbA1c = Glycated hemoglobin, HDL-C = High density lipoprotein cholesterol, LDL-C = Low density lipoprotein cholesterol, VLDL-C = Very low density lipoprotein cholesterol, PAI-1 Ag = Plasminogen activator inhibitor 1 antigen, G = Good glycemic control, P = Poor glycemic control.

[a] *p*-values were calculated using Kruskal-Wallis test and values presented as median (25th-75th percentiles),

[b] Mann-Whitney U test was performed to compare DBP among the groups and values presented as mean ± standard deviation,

* *p*-values<0.05 was considered statistically significant.

was also elevated among the participants who had poor glycemic control [9.35% (8.60–12.43)] relative to those with good glycemic control [6.50% (6.20–6.80)]. The differences in the fasting plasma glucose and the glycated hemoglobin among the participants were significant (*p*< 0.001 vs *p*< 0.001).

The median triglyceride value was raised in the diabetic mellitus patients compared to the normal controls (76.50 (47.50–96.00) mg/dL (Table 2). Among the T2DM participants, those with poor glycemic control had higher triglyceride values [147.50 mg/dL (129.00–180.50)] compared to those with good glycemic control [116.00 mg/dL (80.00–142.00)]. Total cholesterol [196.00 mg/dL (159.75–209.00)], LDL-C [114.80 mg/dL (95.68–136.20)], and VLDL-C [26.00 mg/dL (18.00–28.70)], were all higher among T2DM participants with poor glycemic control compared to those with good glycemic control (total cholesterol [180.00 mg/dL (131.25–205.00)], LDL-C [81.90 mg/dL (53.30–105.00)], and VLDL-C [23.00 mg/dl (19.00–28.00)]). On the other hand, HDL-C was lower among the T2DM participants with poor glycemic control [51.60 mg/dL (38.00–63.00)] compared to those with good glycemic control [64.00 mg/dL (55.00–77.75)]. With the exception of VLDL (*p* = 0.270), the variations observed in the lipid profile among the study participants were statistically significant; triglyceride (*p*< 0.001), total cholesterol (*p* = 0.001), LDL-C (*p*< 0.001), and HDL-C (*p* = 0.002).

T2DM patients with poor glycemic control had higher plasma PAI-1 Ag levels [23.48 ng/mL (21.16–28.74)] compared to both the T2DM subjects with good glycemic controls [22.33 ng/mL (19.72–26.86)] and the normal controls [19.05 ng/ml (17.82–21.30)]. Similarly, PAI-1 activity was enhanced in the T2DM participants with poor glycemic control [4.76 U/mL (4.19–6.02)] than both the participants with good glycemic control [4.49 U/mL (4.06–5.60)] and the control group [4.12 U/mL (3.44–4.45)] (Table 2). The variations observed in the

enzyme among the study participants were statistically significant; PAI-1 Ag ($p < 0.001$), PAI-1 activity level ($p = 0.004$) (Table 2).

## Comparison of PAI-1 activity and PAI-1 Ag level across gender

The mean PAI-1 activity between males and females were not significantly different (5.25 ±3.27 vs 5.25±3.90, $p = 0.998$). Similarly, no significant differences were observed between PAI-1 Ag level in males and females (25.83±16.85 vs 26.42±21.92, $p = 0.888$). With respect to the Type II diabetic group, no significant difference was observed for both PAI-1 activity (5.80 ±3.90 vs 5.80±4.65, $p = 0.996$) and PAI-1 Ag level (28.52±20.13 vs 29.91± 26.15, $p = 0.822$) between males and females. Also, in the control group, PAI-1 activity (4.17±0.63 vs 4.15±0.97, $p = 0.951$) and PAI-1 Ag level (20.44±3.38 vs 19.45 ± 3.36, $p = 0.459$) were not significantly different across the different groups of gender.

## Correlation between Plasminogen Activator Inhibitor-1 antigen level and glycated hemoglobin among the study participants

Fig 1 shows the correlation between PAI-1 Ag and glycated hemoglobin (HbA1c) levels among the study participants. A low positive correlation existed between the enzyme PAI-1 Ag and HbA1c levels of the study participants (r = 0.443, $p < 0.001$).

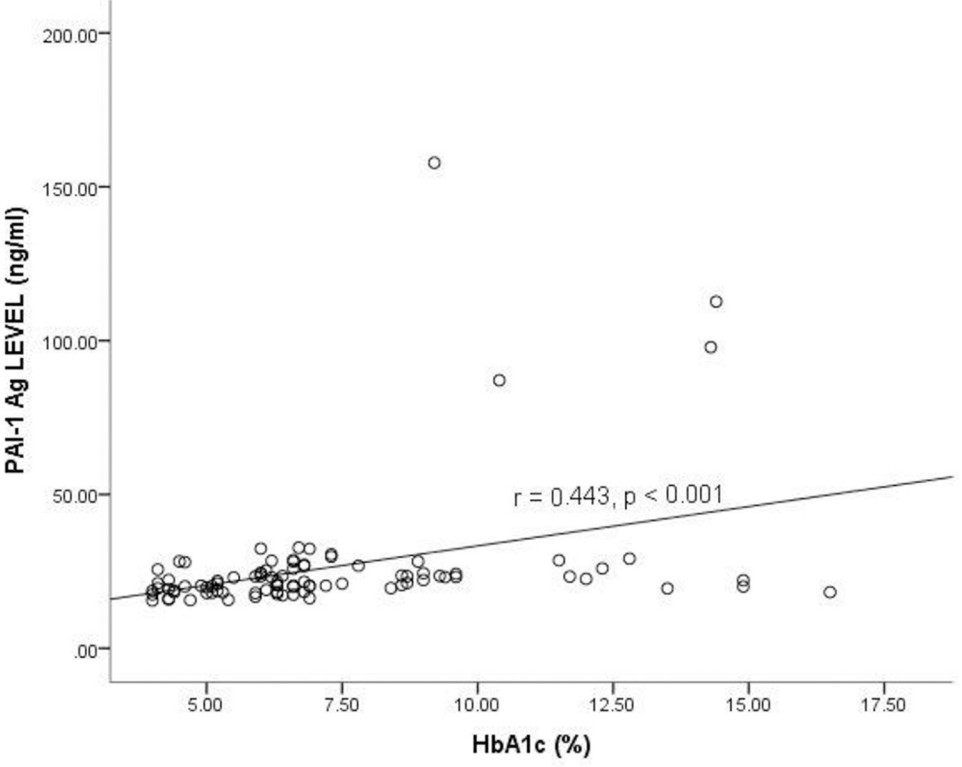

**Fig 1. Correlation between Plasminogen Activator Inhibitor-1 antigen level and glycated hemoglobin among the study participants levels.** HbA1c = Glycated hemoglobin, PAI-1 Ag = Plasminogen activator inhibitor 1 antigen, r = Correlation coefficient. Spearman correlation was used to determine the correlation between the plasma PAI-1 antigen and HbA1c. $p < 0.05$ was considered statistically significant.

## Correlation between Plasminogen Activator Inhibitor-1 activity level and glycated hemoglobin among the study participants

Fig 2 shows the correlation between PAI-1 Activity and glycated hemoglobin (HbA1c) levels among the study participants. A positive correlation was observed between HbA1c and PAI-1 Activity, and it was statistically significant (r = 0.283, $p$ = 0.007).

## Correlation between Plasminogen Activator Inhibitor-1 and fasting blood glucose among the study participants

Fig 3 shows the relationship between PAI-1 and fasting blood glucose (FBG) levels among the study participants. Both PAI-1 antigen and activity levels moderately correlated positively with fasting blood glucose (FBG vs PAI-1 Ag: r = 0.455, $p$ = 0.000 and FBG vs PAI-1 Activity: r = 0.411, $p$ = 0.000) and this is shown in Fig 3.

## Relationship between PAI-1 antigen and BMI, BP, HbA1c, duration of diabetes mellitus and Lipid parameters among the participants

Linear logistic regression models were performed to determine the relationship between PAI-1 antigen and BMI, BP, HbA1c, Duration of Diabetes and Lipid Parameters as shown in (Table 3). PAI-1 Ag levels significantly increased along with increased total cholesterol

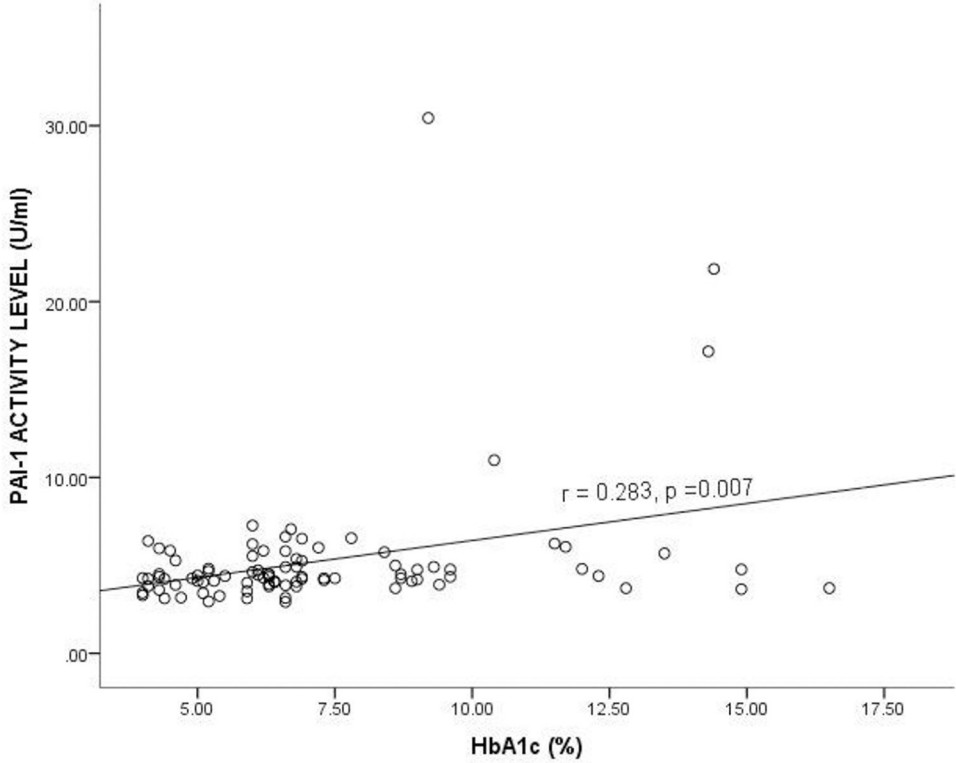

**Fig 2. Correlation between Plasminogen Activator Inhibitor-1 activity level and glycated hemoglobin among the study participants.** HbA1c = Glycated hemoglobin, PAI-1 = Plasminogen activator inhibitor 1, r = Correlation coefficient. Spearman correlation was used to determine the correlation between the plasma PAI-1 activity level and HbA1c. $p<0.05$ was considered statistically significant.

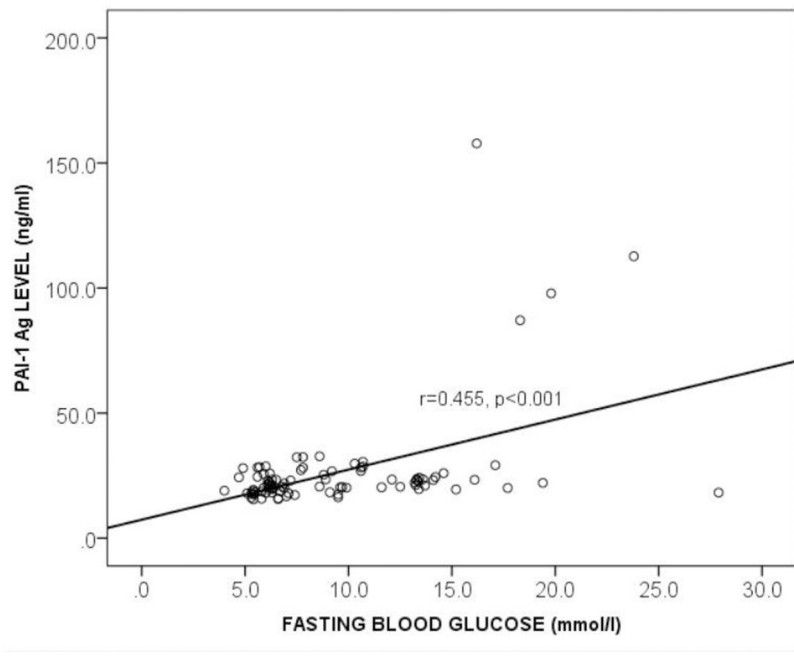

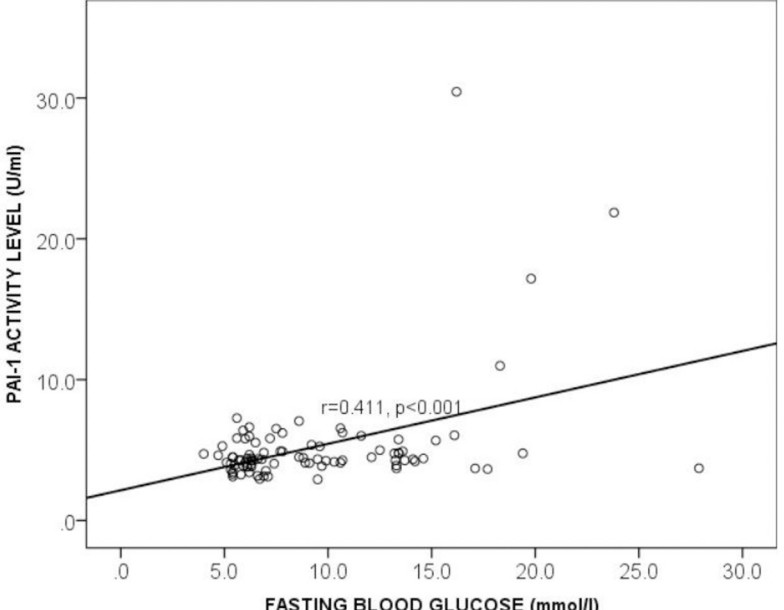

**Fig 3. Correlation between Plasminogen Activator Inhibitor-1 and fasting blood glucose among the study participants levels.** FBG = fasting blood glucose, PAI-1 Ag = Plasminogen activator inhibitor 1 antigen, r = Correlation coefficient. Spearman correlation was used to determine the correlation between the plasma PAI-1 antigen and FBG (Fig 3a) and PAI-1 activity and FBG (Fig 3b). $p < 0.05$ was considered statistically significant.

[$B$ = 0.262 (95% CI: 0.011–0.263), $p$ = 0.033], triglyceride [$B$ = -0.273 (95% CI: -0.223 –-0.009), $p$ = 0.034] and HbA1c [$B$ = 0.419 (95% CI: 1.194–4.759), $p$ = 0.001] (Table 3). However, BMI, blood pressure, HDL-c, LDL-c, VLDL-c and duration of T2DM were not associated with the plasma PAI-1 Ag among the participants as shown in Table 4.

**Table 3. Comparison of PAI-1 activity and PAI-1 Ag across gender.**

| Both groups | Gender | | *p*-value |
|---|---|---|---|
| | **Male** | **Female** | |
| PAI-1 Activity level (U/mL) | 5.25 ± 3.27 | 5.25 ± 3.90 | 0.998 |
| PAI-1 Ag level (ng/mL) | 25.83 ± 16.85 | 26.42 ± 21.92 | 0.888 |
| **T2DM Participants** | | | |
| PAI-1 Activity level (U/mL) | 5.80 ± 3.90 | 5.80 ± 4.65 | 0.996 |
| PAI-1 Ag level (ng/mL) | 28.52 ± 20.13 | 29.91 ± 26.15 | 0.822 |
| **Healthy Controls** | | | |
| PAI-1 Activity level (U/mL) | 4.17 ± 0.63 | 4.15 ± 0.97 | 0.951 |
| PAI-1 Ag level (ng/mL) | 20.44 ± 3.38 | 19.45 ± 3.36 | 0.459 |

Data are presented as Mean ± standard deviation (SD); PAI-1 = Plasminogen activator inhibitor-1; Ag = Antigen

### Relationship between PAI-1 activity levels and BMI, BP, HbA1c, duration of diabetes mellitus and lipid parameters among the participants

Linear logistic regression models were performed to ascertain the relationship between PAI-1 activity level and BMI, BP, HbA1c, Duration of Diabetes and Lipid Parameters as shown in (Table 4). Enhanced PAI-1 activity levels were influenced significantly by increased levels of total cholesterol (B = 0.325 (95% CI: 0.008–0.054), *p* = 0.009), triglyceride (B = -0.262 (95% CI: -0.040 – -0.001), *p* = 0.042), HbA1c (B = 0.389 (95% CI: 0.176–0.829), *p* = 0.003) and VLDL-c (B = -0.227 (95% CI: -0.247 – -0.013), *p* = 0.029) (Table 4). PAI-1 activity levels among the subjects. However, it was not associated with BMI, blood pressure, HDL-c, LDL-c, and duration of T2DM as shown in Table 5.

## Discussion

Globally, type 2 diabetes mellitus (T2DM) kills about 5.1 million people annually [3], and thrombotic events account for approximately 80% of the deaths [14]. The prothrombotic state

**Table 4. Relationship between plasma PAI-1 antigen and BMI, BP, HbA1c, duration of diabetes mellitus and Lipid parameters among the participants.**

| Variables | *B* (95% CI) | S.E | *p*-value[a] |
|---|---|---|---|
| BMI | -0.091 (-1.093–0.385) | 0.371 | 0.343 |
| SBP | -0.177 (-0.407–0.047) | 0.114 | 0.118 |
| DBP | -0.013 (-0.383–0.341) | 0.182 | 0.909 |
| Total Cholesterol | 0.262 (0.011–0.263) | 0.063 | **0.033**[*] |
| Triglyceride | -0.273 (-0.223 – -0.009) | 0.054 | **0.034**[*] |
| HDL-c | 0.136 (-0.081–0.410) | 0.123 | 0.187 |
| LDL-c | 0.109 (-0.084–0.230) | 0.079 | 0.356 |
| VLDL-c | -0.189 (-1.238–0.041) | 0.321 | 0.066 |
| HbA1c | 0.419 (1.194–4.759) | 0.895 | **0.001**[*] |
| Duration of T2DM | 0.095 (-3.232–8.199) | 2.872 | 0.390 |

BMI = Body mass index, SBP = Systolic blood pressure, DBP = Diastolic blood pressure, HDL-c = High density lipoprotein cholesterol, LDL-c = Low density lipoprotein cholesterol, VLDL-c = Very low density lipoprotein cholesterol, HbA1c = Glycated Hemoglobin, T2DM = Type 2 Diabetes Mellitus, *B* = Regression coefficient, CI = Confidence interval, S.E = Standard error.

[a]*p*-values were generated by linear regression models.

[*] *p*-values <0.05 were considered statistically significant.

**Table 5. Relationship between plasma PAI-1 Activity level and BMI, BP, HbA1c, duration of diabetes mellitus and Lipid parameters.**

| Variables | B (95% CI) | S.E | *p*-value[a] |
|---|---|---|---|
| BMI | -0.080 (-0.192–0.079) | 0.068 | 0.408 |
| SBP | -0.152 (-0.070–0.013) | 0.021 | 0.182 |
| DBP | 0.012 (-0.063–0.070) | 0.033 | 0.911 |
| Total Cholesterol | 0.325 (0.008–0.054) | 0.012 | 0.009* |
| Triglyceride | -0.262 (-0.040 –-0.001) | 0.010 | 0.042* |
| HDL-c | 0.135 (-0.015–0.075) | 0.023 | 0.193 |
| LDL-c | 0.116 (-0.015–0.043) | 0.014 | 0.328 |
| VLDL-c | -0.227 (-0.247 –-0.013) | 0.059 | 0.029* |
| HbA1c | 0.389 (0.176–0.829) | 0.164 | 0.003* |
| Duration of T2DM | 0.051 (-0.803–1.289) | 0.525 | 0.645 |

BMI = Body mass index, SBP = Systolic blood pressure, DBP = Diastolic blood pressure, HDL-c = High density lipoprotein cholesterol, LDL-c = Low density lipoprotein cholesterol, VLDL-c = Very low density lipoprotein cholesterol, HbA1c = Glycated Hemoglobin, T2DM = Type 2 Diabetes Mellitus, *B* = Regression coefficient, CI = Confidence interval, S. E = Standard error.

[a]*p*-values were generated by linear regression models.

* *p*-values <0.05 were considered statistically significant.

of the T2DM disease renders affected individuals in a higher risk of dying from cardiovascular events [2, 8, 16, 20]. The hypercoagulable state in these individuals is due to the associated abnormalities in platelet function, enhanced activation of prothrombotic coagulation factors coupled with reduced fibrinolysis [2]. The lowered plasma fibrinolytic activity in the T2DM patients may be ascribable to the elevated plasma antigen and activity levels of plasminogen activator inhibitor-1 (PAI-1) [15]. PAI-1 is a prothrombotic agent produced from endothelial cells, adipocytes, hepatocytes, mononuclear cells and fibroblasts and modulates fibrinolysis by antagonizing tissue plasminogen activator [21]. Good glycemic control during T2DM management has been helpful in reducing the risk of cardiovascular disease onset according to the study by Ida *et al*., (2018). This study was designed to assess the plasma antigen and activity levels of plasminogen activator inhibitor-1 among Ghanaian T2DM patients, with respect to their glycemic controls.

T2DM participants with poor glycemic control expressed the highest concentrations of the plasma PAI-1 antigen compared to those with good glycemic control and the healthy blood donors. Concurrently, the enzyme's activity was greatly enhanced in the T2DM individuals with poor glycemic control relative to those with good glycemic control. The elevated PAI-1 Ag among T2DM participants observed in this study is in accordance with other studies which recorded similar findings [12, 22, 23]. The finding of an enhanced PAI-1 activity level among the T2DM participants in our study is similar to previous studies [13, 15], which agree that the lowered plasma fibrinolytic activity observed in patients with T2DM is mainly related to the elevated plasma antigen and activity levels of the PAI-1. The increased PAI-1 Ag and/or activity levels may be due to the low-grade inflammation caused by the T2DM as the enzyme is known to be an acute-phase protein [24].

During the inflammatory process, there is a significant PAI-1 elevation in response to inflammatory cytokines (TNF-α, IL-1), as observed by McCormack et al., [25] in their *in vitro* study. Inflammatory cytokines are capable of inducing endothelial release of the PAI-1 leading to its elevation in the plasma. T2DM is usually associated with hyperinsulinemia and this hormone has been shown to stimulate transcription of the PAI-1 gene in several tissues

according to the *in vitro* study by El Sayed *et al.*, (2018). Another *in vitro* study confirmed the ability of glucose to stimulate PAI-1 expression in endothelial and vascular smooth muscle cells [26]. The up-regulation of the PAI-1 during the inflammatory event induced by the T2DM may also be influenced by the various polymorphisms associated with the PAI-1 gene as was confirmed by the study which observed much stronger elevation in PAI-1 in the acute-phase after acute-trauma for patients with the 4G-allele than for those with the 5G-allele [24].

Hyperglycemia and hyperinsulinemia are linked with raised levels of plasma PAI-1 levels, and this enlightens eminent quantities of the enzyme in insulin-resistant situations [22]. This study found a positive correlation between HbA1c, FBG and PAI-1 enzyme (both PAI-1 Ag and PAI-1 activity levels) and this is similar to a previous long-term (over 18 years) study between HbA1c and PAI-1 levels [27], directly implicating glycaemia in modulating fibrinolysis potential.

The activity and antigen level of PAI-1 were similar for both males and females across all the three groups studied (Type-2 DM, healthy controls, and the combined group of patient and controls). This gender associated similarity is however, not well understood.

The significant positive correlation identified between HbA1c and the PAI-1 enzyme signifies the influence of hyperglycaemia on the enzyme. Good glycemic control during T2DM management is therefore necessary to reduce the level of PAI-1 in the plasma and this would eventually lower the risk of cardiovascular disease onset in these individuals [15].

In our study, PAI-1 Ag levels significantly increased along with increased total cholesterol, triglyceride, and HbA1c. However, BMI, blood pressure, HDL-c, LDL-c, VLDL-c and duration of T2DM were not associated with the plasma PAI-1 Ag among the participants. Also, enhanced PAI-1 activity levels were influenced significantly by elevated levels of total cholesterol, triglyceride, HbA1c and VLDL-c among the participants, but was not associated with BMI, blood pressure, HDL-c, LDL-c, and duration of T2DM. Hypertriglyceridemia, acute and chronic, were seen to be associated with plasma PAI-1 in a study by Luo *et al.*, [28], and this confirms our finding. Another study involving 10 apparently healthy males showed an elevation in plasma PAI-1 following a 6 hr continuous triglyceride infusion, regardless of the insulin and plasma glucose levels [29]. The study by Krebs *et al.*, (2003) reiterated that the infusion of the triglyceride in the study subjects during their study could trigger vascular tissue release of PAI-1 and this might induce the enzyme's elevation in the plasma, with concomitant enhanced activity level. The variations of the association between triglyceride levels and PAI-1 was observed to be influenced by polymorphisms of the PAI-1 gene, according to a study by Zhang *et al.*, (2014). They concluded in their study that the association between PAI-1 and triglyceride levels showed a steeper slope in participants with the 4G/4G-genotype in patients with coronary artery disease and in type 2 diabetes mellitus [30].

The present study found an association between PAI-1 enzyme and both total cholesterol and LDL-cholesterol and this is similar to a study in 2006 [31]. Again, a recent study also indicated a correlation between PAI-1 activity and LDL-particle size, further doubling the risk of cardiovascular diseases in T2DM [20]. However, other studies have realised inconsistent effects of LDL-cholesterol on plasma PAI-1. According to these studies [32, 33], native-LDL does not stimulate PAI-1 synthesis *in vitro*, unless high concentrations are employed or when LDL is oxidized, or undergoes glycation. An *in vitro* study suggested that; VLDL could trigger a concentration-dependent elevation in PAI-1 expression in endothelial and hepatic cells [34]. The VLDL induction of PAI-1 expression in endothelial cells, according to El Sayed *et al.*, (2018) is mediated via transcriptional activation of the PAI-1 gene. The Hoekstra and co. study again reiterated that apart from the PAI-1 gene transcription, VLDL could also influence the

stability of the PAI-1 mRNA transcripts [34]. Even though our study did not show an association between VLDL-c and PAI-1 Ag in the plasma, the enzyme's activity was enhanced by the increase in the VLDL-c. Plasma PAI-1, both Ag and activity, was not associated with HDL-c in our study and this finding is similar to a previous study [31].

We did not find any association between BMI and PAI-1 levels in our study and this is in support of the study by Bastard *et al.*, [35] which realised that even after weight lowering, PAI-1 expression in human subcutaneous adipocytes was still elevated. The finding from our study is contrary to other studies which observed an association between PAI-1 and obesity [36–38]. The variations in the association between PAI-1 levels and obesity could be linked to the various PAI-1 gene polymorphisms. For instance, a study indicated a perfect association between PAI-1 levels and the 4G/5G-polymorphism in subjects with obesity, but not in lean subjects [39]. Also, BMI was strongly associated with PAI-1 in a study involving Pima Indian subjects with both the 4G/4G and 5G/5G-genotypes, but not in the 4G/5G-genotype [40]. A higher prevalence of obesity was observed in carriers of the 4G-allele than of the 5G-allele in a study which involved 505 humans [41]. Central obesity is therefore regarded as an essential determinant of plasma PAI-1 levels, but the association differs across the genotypes of the 4G/5G-polymorphism. We did not observe any association between PAI-1 and blood pressure as well as the duration of the diabetes mellitus.

The study was limited by our inability to determine the specific polymorphisms of the PAI-1 gene in the T2DM subjects. Also, the ratio of male to female in both subgroups were unbalanced, the non-significant difference may have been masked by the unbalanced gender. To help address this limitation we, recommend that participants be matched for age and gender in future studies.

## Conclusion

PAI-1 antigen/activity is enhanced in poorly controlled Ghanaian T2DM patients. The hypercoagulable state of the affected individuals put them at a higher risk of developing cardiovascular diseases and this contributes to the high mortality rate of the condition. Good glycemic control during T2DM management down-regulates PAI-1 Ag and/or activity. Dyslipidemia influences significantly the plasma antigen and activity levels of PAI-1, further augmenting the development of cardiovascular events in these patients. Markers of fibrinolysis should be periodically assessed during T2DM management. T2DM patients should also be given appropriate anticoagulants in the course of their management to prevent the onset of thrombosis and subsequent cardiovascular events. A further study is recommended to carry out to holistically assess the entire coagulation and fibrinolytic profile of Ghanaian T2DM patients.

## Supporting information

**S1 Questionnaire.**
(DOCX)

**S1 Data.**
(RAR)

## Acknowledgments

We are grateful for the immense contributions of the staff of Nkenkaasu District Hospital and Wenchi Methodist Hospital Laboratories, not forgetting our participants.

## Author Contributions

**Conceptualization:** Charles Nkansah, Otchere Addai-Mensah, Michael Owusu, Alexander Yaw Debrah.

**Data curation:** Charles Nkansah, Kofi Mensah, Felix Osei-Boakye, Samuel K. Appiah, Dorcas Serwaa, Charles A. Derigubah.

**Formal analysis:** Charles Nkansah, Michael Owusu, Richard K. D. Ephraim, Patrick Adu, Felix Osei-Boakye, Dorcas Serwaa, Alexander Yaw Debrah.

**Investigation:** Charles Nkansah, Samuel K. Appiah.

**Methodology:** Charles Nkansah, Otchere Addai-Mensah, Richard K. D. Ephraim, Patrick Adu, Alexander Yaw Debrah.

**Project administration:** Charles Nkansah.

**Resources:** Charles Nkansah.

**Supervision:** Charles Nkansah, Otchere Addai-Mensah.

**Writing – original draft:** Charles Nkansah, Otchere Addai-Mensah, Kofi Mensah, Michael Owusu, Richard K. D. Ephraim, Patrick Adu, Charles A. Derigubah, Alexander Yaw Debrah.

**Writing – review & editing:** Charles Nkansah, Otchere Addai-Mensah, Kofi Mensah, Michael Owusu, Richard K. D. Ephraim, Patrick Adu, Felix Osei-Boakye, Samuel K. Appiah, Dorcas Serwaa, Charles A. Derigubah, Alexander Yaw Debrah.

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
