## [Decision Letter · Decision Letter 0]

1 Dec 2020

PONE-D-20-23161

Plasminogen Activator Inhibitor-1 in poorly controlled vs well controlled Type-2 Diabetes Mellitus patients; a cross-sectional study in a district hospital in Ghana

PLOS ONE

Dear Dr Serwaa

Thank you for submitting your manuscript to PLOS ONE. After careful consideration, we feel that it has merit but does not fully meet PLOS ONE’s publication criteria as it currently stands. Therefore, we invite you to submit a revised version of the manuscript that addresses the points raised during the review process.

We look forward to receiving your revised manuscript.

Kind regards,

Xianwu Cheng, M.D., Ph.D., FAHA

Academic Editor

PLOS ONE

Journal Requirements:

3. We noted that you refer to this study as a cross-sectional study through out the manuscript, but according to your description we would not consider this a cross-sectional study but rather a case control study. In order to avoid confusion we would suggest that you change the wording in your manuscript and avoid referring to this study as a cross-sectional study.

4.Please provide a sample size and power calculation in the Methods, or discuss the reasons for not performing one before study initiation.

Additional Editor Comments (if provided):

None.

Reviewers' comments:

Reviewer's Responses to Questions

**Comments to the Author**

1. Is the manuscript technically sound, and do the data support the conclusions?

Reviewer #1: Yes

Reviewer #2: No

2. Has the statistical analysis been performed appropriately and rigorously? 

Reviewer #1: Yes

Reviewer #2: Yes

3. Have the authors made all data underlying the findings in their manuscript fully available?

Reviewer #1: Yes

Reviewer #2: Yes

4. Is the manuscript presented in an intelligible fashion and written in standard English?

Reviewer #1: Yes

Reviewer #2: Yes

5. Review Comments to the Author

Reviewer #1: I found the research to be rigorous, thorough, reasonable, and well-written. I found the investigation as needed for the patients studied and invites’ other researchers to do the same for their patrons. The research agrees with and builds upon prior research.

The wording and spelling conform with U.S. English minus the few words below. These words are listed for the researchers to confirm the spelling is correct for their intended audience.

U.S. English Your Document

etiology aetiology

Analyzed Analysed

Dyslipidemia Dyslipidaemia

Glycemic Glycaemic

Glycosylated Glycosylatedcated

Hematological Haematological

Hemoglobin Haemoglobin

hemostasis haemostasis

Hyperinsulinemia Hyperinsulinaemia

ischemic ischaemic

neighboring neighbouring

Reviewer #2: The authors provide an interesting and potential important manuscript describing "Plasminogen activator inhibitor-1 in poorly controlled vs well controlled Type-2 diabetes melitus patients; a cross-sectional study in a district hospital in Ghana", The main issues concerning this paper are those concerning the potential associations between PAI-1 and glycaemic.

There are some weak points that need to be addressed by the authors

Major

1. The ratio of male to female in the subgroup is unbalanced, and additional data are needed to explain the gender difference of PAI-1.( see Table1 below)

2. In the table below, abnormal distribution of blood lipids and blood glucose occurred simultaneously between groups. How to determine the relationship between PAI-1 and blood glucose?( see Table2 below)

6. PLOS authors have the option to publish the peer review history of their article (what does this mean?). If published, this will include your full peer review and any attached files.

Reviewer #1: **Yes: **Gerald D. Redwine

Reviewer #2: No

---

## [Author Response · Author response to Decision Letter 0]

5 Dec 2020

Dear Editor,

Thank you for your email. I am pleased to resubmit manuscript number: PONE-D-20-23161 titled “Plasminogen Activator Inhibitor-1 in poorly controlled vs well controlled Type-2 Diabetes Mellitus patients; a case-control study in a district hospital in Ghana" for your consideration.

The concerns raised by the editorial team have been addressed and highlighted below and in the respective sections in the manuscript. I look forward to your favourable response.

Thank you.

 The manuscript meets PLOS ONE’s style requirements.

 The Questionnaire for the study has been uploaded as a supporting document

3. We noted that you refer to this study as a cross-sectional study throughout the manuscript, but according to your description we would not consider this a cross-sectional study but rather a case control study. In order to avoid confusion we would suggest that you change the wording in your manuscript and avoid referring to this study as a cross-sectional study.

 The ‘cross-sectional study’ wording in the entire document has been replaced with ‘case-control study’

4. Please provide a sample size and power calculation in the Methods, or discuss the reasons for not performing one before study initiation.

 The sample size and power calculation has been included in the Methods

 Captions have been included in the Supporting Information files

5. Review Comments to the Author

Reviewer #1: I found the research to be rigorous, thorough, reasonable, and well-written. I found the investigation as needed for the patients studied and invites’ other researchers to do the same for their patrons. The research agrees with and builds upon prior research.

The wording and spelling conform with U.S. English minus the few words below. These words are listed for the researchers to confirm the spelling is correct for their intended audience.

U.S. English Your Document

etiology aetiology

Analyzed Analysed

Dyslipidemia Dyslipidaemia

Glycemic Glycaemic

Glycosylated Glycosylatedcated

Hematological Haematological

Hemoglobin Haemoglobin

hemostasis haemostasis

Hyperinsulinemia Hyperinsulinaemia

ischemic ischaemic

neighboring neighbouring

Changes have been made and all of the above words now conform to U.S. English in the manuscript

Reviewer #2: The authors provide an interesting and potential important manuscript describing "Plasminogen activator inhibitor-1 in poorly controlled vs well controlled Type-2 diabetes melitus patients; a cross-sectional study in a district hospital in Ghana", The main issues concerning this paper are those concerning the potential associations between PAI-1 and glycaemic.

There are some weak points that need to be addressed by the authors

Major

1. The ratio of male to female in the subgroup is unbalanced, and additional data are needed to explain the gender difference of PAI-1 (see Table1 below).

This has been rectified and the gender ratio is balanced now

2. In the table below, abnormal distribution of blood lipids and blood glucose occurred simultaneously between groups. How to determine the relationship between PAI-1 and blood glucose? (see Table2 below).

The relationship between PAI-1 and fasting blood glucose has been established and data embedded in the manuscript.

---

## [Decision Letter · Decision Letter 1]

19 Jan 2021

PONE-D-20-23161R1

Plasminogen Activator Inhibitor-1 in poorly controlled vs well controlled Type-2 Diabetes Mellitus patients; a case-control study in a district hospital in Ghana

PLOS ONE

Dear Dr Serwaa,

Thank you for submitting your manuscript to PLOS ONE. After careful consideration, we feel that it has merit but does not fully meet PLOS ONE’s publication criteria as it currently stands. Therefore, we invite you to submit a revised version of the manuscript that addresses the points raised during the review process.

We look forward to receiving your revised manuscript.

Kind regards,

Xianwu Cheng, M.D., Ph.D., FAHA

Academic Editor

PLOS ONE

Additional Editor Comments (if provided):

Although the authors put effort to somewhat address origial concerns, this editors and the second reviewer that the number of men and women remains the same as before, and other data in table1 have not changed. It is hard to accept the explanation for the previous problem. The author should review the data, re-count and re-statistical analysis it. It'll be final revision time to cosider for publish this paper.

Reviewers' comments:

Reviewer's Responses to Questions

**Comments to the Author**

1. If the authors have adequately addressed your comments raised in a previous round of review and you feel that this manuscript is now acceptable for publication, you may indicate that here to bypass the “Comments to the Author” section, enter your conflict of interest statement in the “Confidential to Editor” section, and submit your "Accept" recommendation.

Reviewer #2: All comments have been addressed

2. Is the manuscript technically sound, and do the data support the conclusions?

Reviewer #2: No

3. Has the statistical analysis been performed appropriately and rigorously? 

Reviewer #2: No

4. Have the authors made all data underlying the findings in their manuscript fully available?

Reviewer #2: Yes

5. Is the manuscript presented in an intelligible fashion and written in standard English?

Reviewer #2: Yes

6. Review Comments to the Author

Reviewer #2: Dear author

Thank you very much for adjusting the gender ratio. However, the number of men and women remains the same as before, and other data in table1 have not changed. I'm sorry that I cannot accept the explanation for the previous problem. The author should review the data, re-count and collate it.

Best Whishes

7. PLOS authors have the option to publish the peer review history of their article (what does this mean?). If published, this will include your full peer review and any attached files.

Reviewer #2: No

---

## [Author Response · Author response to Decision Letter 1]

16 Mar 2021

Dear Editor,

Thank you for your email. I am pleased to resubmit manuscript number: PONE-D-20-23161 titled “Plasminogen Activator Inhibitor-1 in poorly controlled vs well controlled Type-2 Diabetes Mellitus patients; a case-control study in a district hospital in Ghana" for your consideration.

The concern raised by the editorial team have been addressed and highlighted below and in the respective sections in the manuscript. I look forward to your favourable response.

Thank you.

Reviewer’s comment 

Thank you very much for adjusting the gender ratio. However, the number of men and women remains the same as before, and other data in table 1 have not changed. I'm sorry that I cannot accept the explanation for the previous problem. The author should review the data, re-count and collate it.

Authors’ response 

First, the data has been reviewed, recounted and re collated to address the appropriate percentage distribution of the participants’ gender. However, the ratio of male to female in the subgroup still remains unbalanced. It is not possible to balance the gender now, this could have been done at the data collection stage; it cannot be undertaken after data collection. In the methodology we did not indicate anyway where that we controlled for gender (i.e. Collecting equal number of males and females in each arm/sub-group). We therefore have acknowledged this as a limitation of the study. 

Second, additional data for comparison of mean PAI-1 activity and PAI-1 Ag level across Gender was analysed and reported as below. 

Comparison of PAI-1 activity and PAI-1 Ag level across Gender

The mean PAI-1 activity between males and females were not significantly different (5.25±3.27 vs 5.25±3.90, p=0.998). Similarly, no significant differences were observed between PAI-1 Ag level in males and females (25.83±16.85 vs 26.42±21.92, p=0.888). With respect to the Type II diabetic group, no significant difference was observed for both PAI-1 activity (5.80±3.90 vs 5.80±4.65, p=0.996) and PAI-1 Ag level (28.52±20.13 vs 29.91± 26.15, p=0. 822) between males and females. Also, in the control group, PAI-1 activity (4.17±0.63 vs 4.15±0.97, p=0.951) and PAI-1 Ag level (20.44±3.38 vs 19.45 ± 3.36, p=0.459) were not significantly different across the different groups of gender.

Table 3: Comparison of PAI-1 activity and PAI-1 Ag across Gender 

 Gender 

p-value

Both groups Male Female 

PAI-1 Activity level (U/mL) 5.25 ± 3.27 5.25 ± 3.90 0.998

PAI-1 Ag level (ng/mL) 25.83 ± 16.85 26.42 ± 21.92 0.888

T2DM Participants 

PAI-1 Activity level (U/mL) 5.80 ± 3.90 5.80 ± 4.65 0.996

PAI-1 Ag level (ng/mL) 28.52 ± 20.13 29.91 ± 26.15 0.822

Healthy Controls 

PAI-1 Activity level (U/mL) 4.17 ± 0.63 4.15 ± 0.97 0.951

PAI-1 Ag level (ng/mL) 20.44 ± 3.38 19.45 ± 3.36 0.459

Data are presented as Mean ± standard deviation (SD); PAI-1= Plasminogen activator inhibitor-1; Ag= Antigen

Acknowledged limitations of the study.

The study was limited by our inability to determine the specific polymorphisms of the PAI-1 gene in the T2DM subjects. Also, the ratio of male to female in both subgroups were unbalanced, the non-significant difference may have been masked by the unbalanced gender. To help address this limitation we, recommend that participants be matched for age and gender in future studies.

---

## [Decision Letter · Decision Letter 2]

31 Mar 2021

Plasminogen Activator Inhibitor-1 in poorly controlled vs well controlled Type-2 Diabetes Mellitus patients; a case-control study in a district hospital in Ghana

PONE-D-20-23161R2

Dear Dr Serwaa

We’re pleased to inform you that your manuscript has been judged scientifically suitable for publication and will be formally accepted for publication once it meets all outstanding technical requirements.

Kind regards,

Xianwu Cheng, M.D., Ph.D., FAHA

Academic Editor

PLOS ONE

Additional Editor Comments (optional):

Both of original reviewers have indicated acceptance.

Reviewers' comments:

Reviewer's Responses to Questions

**Comments to the Author**

1. If the authors have adequately addressed your comments raised in a previous round of review and you feel that this manuscript is now acceptable for publication, you may indicate that here to bypass the “Comments to the Author” section, enter your conflict of interest statement in the “Confidential to Editor” section, and submit your "Accept" recommendation.

Reviewer #2: All comments have been addressed

2. Is the manuscript technically sound, and do the data support the conclusions?

Reviewer #2: Yes

3. Has the statistical analysis been performed appropriately and rigorously? 

Reviewer #2: (No Response)

4. Have the authors made all data underlying the findings in their manuscript fully available?

Reviewer #2: Yes

5. Is the manuscript presented in an intelligible fashion and written in standard English?

Reviewer #2: Yes

6. Review Comments to the Author

Reviewer #2: Thank you for your explanation. Although there is gender imbalance in the data, there is no significant difference in PAI-1 level among each group, which excludes the influence caused by gender deviation, and the credibility of the results is enhanced to a certain extent.

7. PLOS authors have the option to publish the peer review history of their article (what does this mean?). If published, this will include your full peer review and any attached files.

Reviewer #2: No

---

## [Editor Report · Acceptance letter]

5 Apr 2021

PONE-D-20-23161R2 

Plasminogen Activator Inhibitor-1 in poorly controlled vs well controlled Type-2 Diabetes Mellitus patients; a case-control study in a district hospital in Ghana 

Dear Dr. Serwaa:

I'm pleased to inform you that your manuscript has been deemed suitable for publication in PLOS ONE. Congratulations! Your manuscript is now with our production department. 

Kind regards, 

on behalf of

Associate Prof. Xianwu Cheng 

Academic Editor

PLOS ONE